# Trisomies Reorganize Human 3D Genome

**DOI:** 10.3390/ijms242216044

**Published:** 2023-11-07

**Authors:** Irina V. Zhegalova, Petr A. Vasiluev, Ilya M. Flyamer, Anastasia S. Shtompel, Eugene Glazyrina, Nadezda Shilova, Marina Minzhenkova, Zhanna Markova, Natalia V. Petrova, Erdem B. Dashinimaev, Sergey V. Razin, Sergey V. Ulianov

**Affiliations:** 1Center for Molecular and Cellular Biology, Skolkovo Institute of Science and Technology, 143026 Moscow, Russia; 2A.A. Kharkevich Institute for Information Transmission Problems, Russian Academy of Sciences, 127051 Moscow, Russia; 3Research Centre for Medical Genetics, 115522 Moscow, Russia; 4Friedrich Miescher Institute for Biomedical Research, Maulbeerstrasse 66, 4058 Basel, Switzerland; 5Department of Molecular Biology, Faculty of Biology, M.V. Lomonosov Moscow State University, 119234 Moscow, Russia; 6Laboratory of Structural-Functional Organization of Chromosomes, Institute of Gene Biology, Russian Academy of Sciences, 119334 Moscow, Russia; petrovanv@mail.ru; 7Medical Genetic Center LLC Progen, 117630 Moscow, Russia; 8Center for Precision Genome Editing and Genetic Technologies for Biomedicine, Pirogov Russian National Research Medical University, 117997 Moscow, Russia

**Keywords:** aneuploidy, trisomy, Edwards syndrome, Patau syndrome, chromatin spatial organization, Hi-C, transcription regulation, nucleus structure, nuclear lamina

## Abstract

Trisomy is the presence of one extra copy of an entire chromosome or its part in a cell nucleus. In humans, autosomal trisomies are associated with severe developmental abnormalities leading to embryonic lethality, miscarriage or pronounced deviations of various organs and systems at birth. Trisomies are characterized by alterations in gene expression level, not exclusively on the trisomic chromosome, but throughout the genome. Here, we applied the high-throughput chromosome conformation capture technique (Hi-C) to study chromatin 3D structure in human chorion cells carrying either additional chromosome 13 (Patau syndrome) or chromosome 16 and in cultured fibroblasts with extra chromosome 18 (Edwards syndrome). The presence of extra chromosomes results in systematic changes of contact frequencies between small and large chromosomes. Analyzing the behavior of individual chromosomes, we found that a limited number of chromosomes change their contact patterns stochastically in trisomic cells and that it could be associated with lamina-associated domains (LAD) and gene content. For trisomy 13 and 18, but not for trisomy 16, the proportion of compacted loci on a chromosome is correlated with LAD content. We also found that regions of the genome that become more compact in trisomic cells are enriched in housekeeping genes, indicating a possible decrease in chromatin accessibility and transcription level of these genes. These results provide a framework for understanding the mechanisms of pan-genome transcription dysregulation in trisomies in the context of chromatin spatial organization.

## 1. Introduction

Aneuploidy is a deviation in the copy number of single or several chromosomes. The most common aneuploidies are trisomies (Tr; presence of one extra copy of a certain chromosome) and monosomies (loss of one copy in a pair of homologous chromosomes). However, double trisomies, tetrasomy, pentasomy and even hexasomy of autosomes are also present in some types of cancers [1], multiple copies of large chromosome fragments are found in patients with mental disorders [2] and multiple copies of Y and X chromosome are associated with severe development defects [3]. Autosomal trisomy is the most widespread type of aneuploidy (about 60% of clinical cases) associated with miscarriage [4,5] and abnormalities of embryonic development [6]. Most autosomal trisomies are embryonically lethal [7], but trisomies 13, 18 and 21 are present among newborns with a frequency from 1:5000 to 1:1000 [8,9] and are manifested in Patau, Edwards and Down syndromes, respectively. These disorders are characterized by some common symptoms such as: intellectual disability, heart defects, multiple congenital malformations and facial dysmorphias. Pathology of trisomies has been initially associated with the presence of extra copies of genes located on the supernumerary chromosome and, as a consequence, increased expression of these genes. For instance, Down syndrome phenotype has been associated with the so-called Down Syndrome Critical Region located at the long arm of chr21. Similarly, regions of chromosome 18 critical for Edrwards syndrome have been proposed in genetic studies [10]. However, later observations challenged the “gene dosage” hypothesis, showing that the overexpression of a limited number of genes is not sufficient for the manifestation of all symptoms presented in individuals with supernumerary chromosomes [11,12]. As an alternative, the concept of disrupted cellular homeostasis has been proposed [13]. This model postulates that the pathogenicity of trisomies is caused not by the gain of a particular extra chromosome but by the presence of an extra chromosome per se. One of the characteristic features of trisomies is a pan-genomic transcriptional dysregulation that could be a driver of homeostasis disruption. Shifts in the gene expression program, in turn, could be caused by an increased expression of regulatory proteins encoded on the trisomic chromosome [14]. Their activity could change profiles of epigenetic marks [15], chromatin accessibility [16] and DNA methylation profiles [17,18] across the genome, modulating activity of gene promoters. Another possible reason for trisomy-induced transcription dysregulation is reorganization of the genome spatial structure [19] and distal contacts between regulatory elements in cis and in trans [20]. Indeed, previous studies revealed that the presence of an extra chromosome induces changes in the structure of the 3D genome at multiple levels. In fibroblasts carrying a third copy of chromosome 21 (Tr21), several other chromosomes change their compaction and radial positions within the nucleus [21]. In Tr21 induced pluripotent stem cells (iPSCs) and iPSC-derived neural progenitor cells (NPCs), global changes in the pattern of interchromosomal, inter-TAD (topologically associated domains) and looping contacts were observed [22]. Cultured human colonic epithelial cells (HCEC) carrying an extra copy of chromosome 7 exhibit local differences in A/B chromatin compartment profiles accompanied by changes in transcription level within loci whose compartment state is switched between normal and trisomic cells [23]. Even the presence of a small supernumerary marker chromosome (sSMC) has been reported to influence the arrangement of other chromosomes within the cell nucleus [24], even though sSMCs tend to colocalize with their corresponding sister chromosomes [25]. The open question is whether different trisomies exhibit some common features in 3D genome reorganization.

## 2. Results

### 2.1. Trisomies Affect Contacts between Large and Small Chromosomes

To systematically investigate the impact of trisomies (Tr) on chromatin spatial organization, we performed Hi-C on chorion cells carrying additional chromosome 13 (Patau syndrome, Tr13) or chromosome 16 (Tr16, the most frequent trisomy in the analysis of spontaneous abortion material) and on cultured primary skin fibroblasts with a third copy of chromosome 18 (Edwards syndrome, Tr18). The presence of extra chromosomes was revealed by the array comparative genome hybridization (aCGH) technique for Tr13 and Tr16 and by chromosomal analysis and fluorescence in situ hybridization (FISH) with whole-chromosome probes for Tr18 (Appendix A). As a control, we used chorion cells without genomic imbalances on aCGH and fibroblasts from a donor with normal G-banding karyotype. Hi-C libraries were prepared following s previously published protocol [26] using the DpnII restriction enzyme for chromatin fragmentation. Hi-C experiments were performed in at least two independent technical replicates. Chorion cells without genomic imbalances detectable by CGH-array hybridization were isolated from two male and three female donors, and Tr13 and Tr16 chorion were obtained from one female and one male donor. Primary Tr18 and normal fibroblasts were isolated from female donors (one donor for the Tr18 and one donor for the normal fibroblasts). All comparisons between Tr and normal cells were performed within the same genetic background (XX or XY). Hi-C libraries were sequenced with 11.5–150 million paired-end reads, and 6.1–93.3 million unique contacts were retrieved after data processing with *distiller* (see Section 4 and Appendix A), which allowed us to analyze chromatin contact profiles at up to 50 kb resolution. Notably, aCGH does not discriminate free trisomy from the translocation one. It is of particular importance for Tr13 because chr13 is frequently involved in Robertsonian translocations. However, translocation between chr13 and other chromosomes would inevitably lead to a drastic increase in contact probability between the two chromosomes. We do not detect abnormally high interaction frequency between chr13 and any other chromosome in Hi-C data from Tr13 chorion (see Figure 1B). This clearly indicates that Tr13 cells used in this study carry free extra chr13.

First, we found that the presence of extra chromosomes does not cause substantial changes in chromatin folding within chromosome territories, which was revealed by analyzing the dependence of the contact probability on the genomic distance *P_c_*(*s*) in normal and Tr cells (Figure 1A). In addition, the typical shape of *P_c_*(*s*) curves together with an assessment of the statistics describing the proportion of different types of ligation products (non-ligated DNA ends, self-circles; Appendix A) indicate that Hi-C data are suitable for downstream examination.

Analysis of pairwise interactions between chromosomes in normal cells revealed that small chromosomes form a contact cluster and show a low interaction frequency with large chromosomes, which form a less pronounced but noticeable cluster in both chorion cells and fibroblasts (Figure 1B). Therefore, we arbitrary defined chromosomes 16–22 as “small” and chromosomes 1–15, X as “large”. These observations are consistent with previous reports on an increased frequency of interactions between small chromosomes [27]. This could be explained by distinct localization of small and large chromosomes within the nucleus: while large chromosomes tend to be located at the nuclear periphery and contact with the lamina, small chromosomes are preferentially localized at the nuclear interior [28]. It should be noted that in chorion cells, gene-poor chr18 interacts at a relatively high frequency only with chr20 and chr21, which contain fewer genes compared to other small chromosomes (see below, Figure 2F). In Tr18 fibroblasts, chr18 displays an abnormally strong contact with chr9 (Figure 1B, bottom left panel). As revealed by visual inspection of the Hi-C maps (Appendix A), this is likely caused by a local rearrangement between 18q and the pericentromeric region of 9q. FISH imaging of chr9 with subtelomeric probes did not show the presence of split chr9 since both probes are readily detected within the same chromosome (Appendix A). This suggests that the rearrangement did not result in the formation of a chimeric chromosome between chr9 and chr18. Nonetheless, chr9 was excluded from all downstream comparisons of normal and Tr18 fibroblasts.

The pattern of interchromosomal contacts is generally preserved in Tr13, Tr16 and Tr18 (compare left and middle panels in Figure 1B). However, we found that the presence of extra chromosomes induces moderate yet statistically significant and opposite changes in contacts between small and large chromosomes (Figure 1C). In Tr13 and Tr18, we observed further segregation of the large and small chromosome clusters accompanied by an increase in contact frequency between small chromosomes, while Tr16 cells are characterized by a partial intermingling of the clusters and a pronounced decrease in interactions of small chromosomes. Importantly, variations in chromosome contact patterns in pairwise comparisons of normal cells from different donors are remarkably less pronounced (especially for the “small” cluster) yet statistically significant in some pairs (Appendix A). We thus concluded that the presence of an extra chromosome in the nucleus can systematically influence frequency of contacts between other chromosomes and that it exceeds the degree of variability between individual donors.

### 2.2. Individual Chromosomes Respond Differentially to the Presence of an Extra Chromosome

To track the behavior of individual chromosomes in normal and Tr cells, we introduce the term “chromosome entourage” (CE), which describes a spectrum of contacts of each chromosome with all other chromosomes. Within the CE, a chromosome has “close” and “distal” partners, i.e., chromosomes interacting with the given chromosome with high and low frequency, respectively. As an example, for chr22, all small chromosomes (except chr18) are close partners, while for chrX, large chromosomes are close partners (Figure 1B). For each chromosome, we calculated a Spearman’s correlation coefficient *r* between its contact frequencies with other chromosomes in normal cells and fold change of contact frequencies (FCCF) in Tr cells. If *r* > 0 (statistically significant), then the CE of a given chromosome did not qualitatively change in Tr cells compared to cells with normal karyotype. In other words, in Tr cells, the chromosome interacts more intensively with its close partners from normal cells and reduces contacts with distal partners (Figure 2A). *r* < 0 (statistically significant) suggests an increase of contact frequency with chromosomes that were distal partners of a given chromosome in normal cells (a tendency to CE “reversal”). Finally, *r* equal or close to 0 suggests that changes of the chromosome CE are stochastic, i.e., spatial interactions of the chromosome in trisomy change irrespective to its contact pattern in normal cells.

In all trisomies, we found that *r* varies considerably between different chromosomes regardless of their size and the frequency of contacts in normal cells with the chromosome that is present in three copies in Tr cells (representative examples are shown in Appendix A). Thus, chromosome CEs respond individually to the presence of extra chromosomes. This is also true for the extra chromosomes in Tr cells (Figure 2B). To systematically investigate the tendencies of chromosome CE changes in trisomies, we plotted the *r* for all chromosomes and ranked it by the value and statistical significance (Figure 2C). Interestingly, in Tr13 and Tr18, *r* > 0 (or close to 0) for all chromosomes, whereas in Tr16, *r* < 0 (or close to 0) for all chromosomes, except for chr14 and chr15. In addition, in Tr13 and Tr18, the chromosomes presented in triplicate showed intermediate *r* values comparing to other chromosomes. In Tr16, chr16 demonstrated the most significant negative *r* value (*r* = −0.87, *p =* 1.3 × 10^−7^*)*. When comparing normal cells from different donors, the *r* values for the majority of chromosomes were not statistically significant (Appendix A). It is important to note that an *r* that is not statistically significant or close to 0 might reflect either considerable but stochastic changes in the CE or absence of any detectable changes. We assumed that these two scenarios could be discriminated by the assessment of FCCF dispersion: a high value indicates that the chromosome remarkably changes interaction frequency with at least some partners in Tr and a low value indicates noisy fluctuations in the contact profile. As a baseline (expected) level, we used the third quartile of the FCCF dispersion derived from pair-wise comparisons of normal chorion cells (Figure 2D). Applying this threshold, we determined the following chromosomes exhibiting stochastic yet remarkable changes of CE (FCCF dispersion is higher than expected): chr19 in Tr13; chr18 and chr21 in Tr16; chr18, chr20 and chr21 in Tr18 (Figure 2E).

A surprisingly short list of chromosomes with stochastic perturbations in the CE and the fact that trisomies studied in different cell types (Tr13 in chorion and Tr18 in fibroblasts) share chr18 and chr21 made us ask what properties are common for chromosomes in this list. Chr18 and chr21 contain the lowest numbers of protein-coding genes (coding sequences (CDSs) represent only 0.7% and 0.78% of their length, respectively; Figure 2F). In addition, chr18, chr20 and chr21 possess the highest degree of association with the nuclear lamina within the “small” cluster, as revealed by plotting the coverage of previously identified lamina-associated domains conserved among a broad spectrum of non-related cell types (cLADs [29]; Figure 2G). In contrast, chr19 exhibits the highest coverage with CDSs (3.96%) and the lowest content of cLADs (0.3%). Thus, LAD and gene content may be among the determinants of the chromosome behavior in the presence of extra copies of chromosomes.

To verify our Hi-C-based observations, we visualized nuclear localization of chr18, chr21 and chr22 in normal and Tr18 cultured primary fibroblasts using FISH with whole-chromosome probes. These chromosomes were selected because they demonstrated different behavior in the Tr18 cells (Figure 2C). In agreement with previous studies (reviewed in [28]), we observed a difference in radial positions of these chromosomes in normal cells (Figure 3A): heterochromatin-rich chr18 is located close to the nuclear lamina, while euchromatic chr21 and chr22 reside more centrally. In Tr18 cells, chr18 shifts towards the nucleus center, while chr21 moves to the nuclear periphery, and chr22 does not change its radial position significantly (Figure 3B). This result supports the Hi-C data indicating that chr18 and chr21, but not chr22, change their CE in Tr18 fibroblasts.

### 2.3. Trisomies Alter Chromatin Compaction throughout the Entire Genome

To track changes in the local chromatin compaction in Tr cells, we analyzed Hi-C data at 50 kb resolution. For each genomic bin in normal and Tr cells, we calculated the ratio *R* between the number of contacts within a 1 Mb vicinity and the sum of all cis-contacts (i.e., within the harboring chromosome; Figure 4A). The higher the *R*, the higher chromatin density of the region containing the given genomic bin. Next, we determined bins with increased and decreased *R* in Tr cells as compared to normal and combined nearby bins of the same type, obtaining a list of genomic regions with altered compaction in Tr cells (Figure 4B; see Section 4 for the details). These regions vary from 100 kb to 1.4 Mb with median length of 150 kb for both compacted and decompacted loci in all trisomies. In Tr13 and Tr18, the proportion of regions with altered compaction is significantly higher on large chromosomes as compared to small ones (Tr13: *p* = 0.006, Tr18: *p* = 0.0005 in a Mann–Whitney U test), while in Tr16, both clusters are affected similarly (Figure 4C). In the Tr13 and Tr18 cells, we found a strong correlation between the cLAD content and the proportion of compacted regions on large, but not small, chromosomes (Figure 4D). We did not observe any statistically significant trends for the decompacted regions. At first glance, our observations suggest that Tr13 and Tr18, but not Tr16, induce chromatin compaction predominantly in cLADs. However, comparable proportions of compacted (as well as decompacted) regions are localized both inside and outside cLADs (Figure 4E). Assuming that the cLADs could not fully correspond to LADs in the analyzed fibroblasts and chorion cells, we repeated the analysis using the A/B compartment profile derived from our Hi-C data (Appendix A) instead of the cLAD positions. Again, compacted regions are almost equally distributed between A and B compartments. However, for decompacted regions, we observed a remarkable disproportion, especially in the Tr18 cells, where 90% of decompacted loci are fully located in the B compartment (Figure 4F). This suggests that all studied trisomies induce partial decondensation of chromatin predominantly in the repressed genome loci. Consistently, protein-coding genes are mostly underrepresented in decompacted regions of large and small chromosomes and distributed randomly relative to the compacted loci. The exception is Tr16, where protein-coding genes are moderately enriched in compacted regions (Figure 4G, Appendix A). Interestingly, a cohort of genes ubiquitously expressed across different cell types and essential for cellular maintenance (housekeeping genes) is slightly enriched in the compacted loci in Tr18 cells (Figure 4H, Appendix A) that potentially suggests their downregulation.

### 2.4. Chromosome Behavior in Tr21 Cells

Finally, to expand our observations on other trisomies and cell types, we reanalyzed previously published Hi-C data [22] from induced pluripotent stem cells (iPSCs) and iPSC-derived neuronal progenitor cells (NPCs) carrying additional chromosome 21 (Tr21; Down syndrome). We found that the presence of an extra chr21 induces a relatively weak yet statistically significant (*p <* 0.0001 in a Mann–Whitney U-test) drop in contact frequency between small chromosomes in Tr21 iPSCs (Figure 5A), similar to Tr16 (Figure 1A). Intriguingly, this is reverted in Tr21 NPCs differentiated from the Tr21 iPSCs: in these cells, small chromosomes interact with each other more frequently as compared to euploid NPCs. In addition, chromosome clusters become slightly more segregated in the Tr21 NPCs (gray boxplots in Figure 5A). This suggests that the presence of the same extra chromosome might have different effects on the large-scale nucleus structure in different cell types. For chr21, we observed a trend of CE “reversal” in the Tr21 iPSCs and of CE “enforcement” in the Tr21 NPCs (Figure 5B). This is also true for almost all other small chromosomes, except for chr18 in Tr21 NPCs, which does not change contact pattern significantly (Figure 5C, right panel). Thus, chr18 is the only chromosome from the “small” cluster whose interactions do not change or change stochastically in three out of four analyzed trisomies (Tr16, Tr18 and Tr21) in unrelated cell types (chorionic cells, fibroblasts and NPCs, respectively). Among large chromosomes, only chr1, chr11 and chr14 do not follow the common trend of CE “enforcement” in Tr21 NPCs. We note that these chromosomes exhibit the same behavior in Tr21 iPSCs as well (Figure 5C, left panel). This suggests that there should be a mechanism allowing some chromosomes to escape shifts in contact patterns during cell differentiation.

## 3. Discussion

Chromatin occupies about 30% of the cell nucleus volume in human cells [30]. This and the fact that the DNA content of the nucleus naturally doubles upon replication suggest that extra chromosomes (even the largest ones) could not be a simple mechanical obstacle for the nucleus formation and functioning. Indeed, autosomal aneuploidies including the presence of multiple copies of one chromosome or an entire set of chromosomes are easily tolerated by cells in culture while are typically hazardous or fatal for the organism. One reason (though probably not the only one) for that is pan-genomic transcriptional dysregulation, which is observed in all trisomies [31,32,33]. In this regard, an extra chromosome should be considered not as just an inert DNA mass localized somewhere inside the nucleus, or as just a source of additional copies of genes, but as a “bull in a China shop”, whose presence disturbs some fine-tuned structural and functional relationships between (and within) chromosomes. What relationships can be affected by an extra chromosome?

The lamina–nucleolus axis is one of the determinants of the nuclear structure since chromosomes are attached to both entities via nucleolus-associated and lamina-associated chromatin domains [34]. In an individual nucleus, only about 30% of LADs are associated with the lamina [35]. Thus, chromosomes containing a large proportion of LADs (predominantly chromosomes from the large cluster) may compete for localization at the nuclear periphery, which is important for the maintenance of a repressed state of genes located in LADs [36]. An extra chromosome could act as an additional player in this competition, decreasing the probability of contact with the lamina for other chromosomes. In this case, the higher the LAD content of the extra chromosome, the stronger its influence on the chromatin–lamina interactome and the more pronounced the changes in gene expression, which might be particularly critical at early development when the organ rudiments are formed. This potentially explains why extra copies of large chromosomes are almost always embryonically lethal and why trisomies result in multiple development abnormalities. If a chromosome has lost the competition and thus is not attached to the lamina, it inevitably affects the contacts (and localization) of other chromosomes because “the loser” chromosome is wrongly placed within the nucleus. Previously, it has been shown that the presence of an extra chr21 in chorion cells alters radial positions of chr1 and chr3 and induces compactization of chr1 and chr17 [21]. This is in line with our observations on alterations of contact profiles between almost all chromosomes in trisomic cells. Moreover, we found that specifically Tr13 and Tr18, but not Tr16, are characterized by a strong dependency between the proportion of loci with altered compaction and LAD content of a chromosome. Since LAD coverage of chr13 and chr18 is very similar (and about three-fold higher than that for chr16), this suggests disturbed chromatin–lamina interactions as a potential reason for the changes in chromatin compaction within chromosomes. Absence of detectable changes in LAD profiles previously observed in Tr21 fibroblasts [15] is also understandable within this concept: the total LAD length of chr21 is small (Figure 2G, blue bars) and therefore potentially insufficient to successfully compete with other chromosomes.

It should be emphasized that the disturbances in the profiles of interchromosomal contacts in cells with aneuploidies discussed above are only one of the factors that can lead to pathological consequences. Another factor that has long been appreciated is the presence of additional copies of genes, whose expression level is not compensated by any special mechanisms, as in the case of sex chromosomes. Large chromosomes harbor a significant number of genes, although the gene density in large chromosomes is lower than in gene-rich small ones. The large number of genes on large chromosomes may be the main cause of embryonic lethality of large chromosome trisomies. The significance of gene imbalance is well illustrated by the situation in the cluster of small chromosomes. These chromosomes are typically gene-rich (apart from chr18 and chr21). Thus, gene disbalance per se caused by the presence of such chromosomes likely leads to the pathological consequences of trisomies by disturbing regulatory networks at early development. This would explain why the majority of trisomies of small chromosomes are embryonically lethal. However, it should be noted that gene-rich chromosomes tend to be located at the nuclear interior [37]. It is relevant to assume that the presence of these extra chromosomes should primarily affect other small chromosomes. Indeed, the proportion of loci with altered compaction on small chromosomes is the highest in Tr16 compared to Tr13 and Tr18. In addition, we found that extra chr16 significantly decreases contact frequency within the “small” cluster of chorion cells (up to 20% for certain chromosome pairs, Figure 1C). It is attractive to suppose that extra small chromosomes compete with their counterparts for contacts with each other and with the nuclear speckles and impair pair-wise interactions within the “small” cluster in this way. However, extra chr21 in Tr21 NPCs—in contrast—increases the contact frequency within the “small” cluster. Thus, extra small chromosomes may influence chromatin contact patterns differentially, and moreover, their effects could be cell type-specific, as seen in the comparison of Tr21 iPSCs and Tr21 NPCs (Figure 5). A large number of active genes in these chromosomes and, as a consequence, a high density of activatory condensates [38] may potentially provide the ability of such chromosomes to interrupt trans regulatory networks established between other chromosomes.

The most intriguing observation is that some chromosomes change their contact patterns in a stochastic manner in Tr cells and that such behavior of a certain chromosome could be observed in non-related cell types with different extra chromosomes (chr18 in Tr16 chorionic cells, Tr18 fibroblasts and Tr21 NPCs). We assume that such chromosomes establish different contact spectrums in distinct subpopulations of Tr cells. In other words, a “stochastic” behavior in bulk Hi-C data potentially reflects superposition of several states. This could be dictated by factors other than the presence of an extra chromosome per se, such as a degree of cellular senescence (which remarkably impacts chromatin 3D structure [39,40,41]) and differentiation state. The last is of particular importance in a developing embryo and potentially contributes to the enigmatic picture of somatic abnormalities in newborns with trisomies when multiple organs and systems are affected, but to a different extent. The question is why some chromosomes change their contact pattern “systematically” (for example, decreasing interactions with the “small” cluster and increasing with the “large” one), while others do it stochastically? Our data suggest that a combination of the chromosome size, LAD coverage and gene density could determine chromosome behavior in trisomic cells. However, the contribution of each factor requires further studies on large cohorts of donors with normal and trisomic genetic background.

## 4. Materials and Methods

### 4.1. Biological Materials

Human primary fibroblasts were obtained from the collection of the Research Centre for Medical Genetics (Moscow, Russia) and the Center for Precision Genome Editing and Genetic Technologies for Biomedicine, Pirogov Russian National Research Medical University (Moscow, Russia). Fibroblasts were grown in Dulbecco’s Modified Eagle’s Medium (DMEM), supplemented with 10% fetal calf serum, at 37 °C and 5% of CO_2_.

Chorion cells were isolated from abortion tissues in the Medical Genetic Center LLC Progen (Moscow, Russia). Chorionic villi were cut with sterile scissors, and unicellular suspension was obtained by incubating the sample with collagenase I (1 mg/mL) in DMEM, supplemented with 10% fetal calf serum, at 37 °C for 40 min.

### 4.2. Microarray Analysis

For the determination of chromosome copy number, fetal DNA was extracted from abortion tissues using reagents kits «DNA-Extran2» from Syntol (Moscow, Russia), labeled with fluorescent dyes and hybridized on the microarray Agilent GenetiSure PreScreen Array 60 K (Agilent Technologies, Santa Clara, USA) according to Agilent’s protocol. An Agilent SureScan Microarray Scanner was used for the microarray preliminating screening. Interpretation of the results was performed in Agilent CytoGenomics soft.

### 4.3. Hi-C Library Preparation

Hi-C libraries were prepared as described previously [26], with minor modifications. A total of 5–10 million fibroblasts or chorion cells were fixed in 1× PBS containing 2% formaldehyde (Sigma-Aldrich, Burlington, VT, USA) for 10 min, with occasional mixing. The reaction was quenched by 125 mM glycine (Sigma-Aldrich). Cells were pelleted by centrifugation (1000× *g*, 10 min, 4 °C), resuspended in 50 μL 1× PBS, snap-frozen in liquid nitrogen and stored at −80 °C. Cells were lysed in 1.5 mL isotonic buffer (50 mM Tris-HCl pH 8.0 (Sigma-Aldrich), 150 mM NaCl (Sigma-Aldrich), 0.5% (*v*/*v*) NP-40 substitute (Sigma-Aldrich), 1% (*v*/*v*) Triton-X100 (Sigma-Aldrich), 1× Halt™ Protease Inhibitor Cocktail (Thermo Scientific, Waltham, MA, USA)) on ice for 15 min. Cells were harvested by centrifugation at 2500× *g* for 5 min, resuspended in 100 μL 1× DpnII buffer (New England Biolabs, Ipswich, MA, USA) and harvested again. The pellet was resuspended in 200 μL 0.3% SDS (Sigma-Aldrich) in 1.1× DpnII buffer (New England Biolabs) and incubated at 37 °C for 1 h. Then, 330 μL 1.1× DpnII buffer and 53 μL 20% Triton X-100 were added, and the cells was incubated at 37 °C for 1 h. A total of 600 U DpnII enzyme (New England Biolabs) was added, and the chromatin was digested overnight (14–16 h), at 37 °C, with shaking (1400 rpm). In the morning, 200 U DpnII enzyme was added, and the cells were incubated for an additional 2 h. DpnII was then inactivated by incubation at 65 °C for 20 min. The nuclei were pelleted for 10 min at 5000× *g*, washed with 100 μL 1× NEBuffer 2 (New England Biolabs) and resuspended in 125 μL 1.2× NEBuffer 2. Cohesive DNA ends were biotinylated at 37 °C for 75 min by adding 25 μL biotin fill-in mixture (0.025 mM dATP (Thermo Scientific), 0.025 mM dGTP (Thermo Scientific), 0.025 mM dTTP (Thermo Scientific), 0.025 mM biotin-14-dCTP (Thermo Scientific), 0.8 U/μL Klenow enzyme (New England Biolabs)). Nuclei were pelleted at 3000× *g* for 5 min and washed once with 300 μL 1× T4 DNA ligase buffer (Thermo Scientific). Chromatin fragments were ligated at 20 °C for 6 h in the presence of 75 U T4 DNA ligase (Thermo Scientific). The cross-links were then reversed by overnight incubation at 65 °C in the presence of proteinase K (100 μg/mL) (Sigma-Aldrich). Next, the DNA was purified by single phenol-chloroform extraction and ethanol precipitation. To remove residual RNA, samples were treated with 50 μg of RNase A (Thermo Scientific) for 45 min at 37 °C. The DNA was additionally purified using Agencourt AMPure XP beads (Beckman Coulter, Brea, CA, USA). Biotinylated nucleotides from the non-ligated DNA ends were removed by T4 DNA polymerase (New England Biolabs) in NEBuffer 2, supplemented with 0.025 mM dATP and 0.025 mM dGTP, at 20 °C for 4 h. Next, the DNA was purified using Agencourt AMPure XP beads. The DNA was then sheared to a size of approximately 100–1000 bp in the sonication buffer (50 mM Tris-HCl, pH 8.0, 10 mM EDTA, 0.1% SDS) using a VirSonic 100 (SP Scientific, Warminster, USA). The samples were concentrated and purified using AMICON Ultra Centrifugal Filter Units (Merck, Darmstadt, Germany) to a total volume of approximately 50 μL. The DNA ends were repaired by adding 62.5 μL MQ water, 14 μL 10× T4 DNA ligase reaction buffer, 3.5 μL 10 mM dNTP mix (Thermo Scientific), 5 μL 3 U/μL T4 DNA polymerase (New England Biolabs), 5 μL 10 U/μL T4 polynucleotide kinase (New England Biolabs) and 1 μL 5 U/μL Klenow DNA polymerase (New England Biolabs) and incubating at 20 °C for 30 min. After the DNA purification with Agencourt AMPure XP beads, an A-tailing reaction was performed by adding 6 μL 10× NEBuffer 2, 1.2 μL 10 mM dATP, 3.6 μL 5 U/μL Klenow (exo-) (New England Biolabs) and MQ water to a total volume of 60 μL. The samples were incubated for 30 min at 37 °C in a PCR machine, and the enzyme was then heat-inactivated at 65 °C for 20 min. The DNA was purified using Agencourt AMPure XP beads and eluted with 200 μL 10 mM Tris-HCl (pH 8.0). Biotin pulldown of the ligation junctions was performed as described previously [26]. The washed beads attached to captured ligation junctions were resuspended in 50 μL adapter ligation mixture, composed of 41.5 μL MQ water, 5 μL 10× T4 DNA ligase reaction buffer (Thermo Scientific), 2.5 μL Illumina TruSeq adapters and 1 μL 5 U/μL T4 DNA ligase (Thermo Scientific). Adapter ligation was performed at 22 °C, for 2.5 h, and the beads were washed twice with 100 μL TWB (5 mM Tris-HCl, pH 8.0, 0.5 mM EDTA, 1 M NaCl, 0.05% Tween-20 (Sigma-Aldrich)), once with 100 μL 1× binding buffer (10 mM Tris-HCl, pH 8.0, 1 mM EDTA, 2 M NaCl) and once with 100 μL CWB (10 mM Tris-HCl, pH 8.0 and 50 mM NaCl) and then resuspended in 20 μL MQ water. Test PCR reactions containing 4 μL streptavidin-bound Hi-C library were performed to determine the optimal number of PCR cycles. The PCR reactions were performed using KAPA High Fidelity DNA Polymerase (KAPA Biosystems, Merck, Darmstadt, Germany) and Illumina PE1.0 and PE2.0 PCR primers (10 pmol each). Four preparative PCR reactions were performed for each sample. The PCR mixtures were combined, and the products were purified using Agencourt AMPure XP beads.

### 4.4. Hi-C Data Analysis

Raw reads were processed using the distiller-nf pipeline (https://github.com/open2c/distiller-nf (accessed 1 on June 2023)) with a project file available here: https://gist.github.com/Phlya/b1cdceb8124d787731e654a7edaedb82 (accessed on 10 July 2023).

Briefly, reads were mapped using *bwa mem* to the hg38 reference genome. Resulting .bam files were parsed using *pairtools parse* into pairs, which were deduplicated with max_mismatch_bp = 1. Deduplicated pairs with mapq ≥30 were binned using *cooler cload pairs* to create 1000 bp resolution cooler files. All datasets were subsampled to the same number of reads using *cooltools random-sample*. Then, they were coarsegrained to multiple resolutions using *cooler zoomify*. All resolutions were then balanced to create three different weights: genome-wide, cis-only and trans-only. Depending on the analysis, cis-only or trans-only weights were used.

The resulting .mcool files were used in the quaich pipeline, which used *cooltools* for analysis of Hi-C data (https://github.com/open2c/quaich accessed on 1 June 2023) with a config file available here: https://gist.github.com/Phlya/9f461871c5dd97015fa51b5b9e83c3d2 accessed on 10 July 2023).

The results of the pipeline included the cis- and trans- expected, which were used to plot *P_c_*(*s*) curves and average inter-chromosome interaction frequencies, respectively. In addition, we used the compartment annotation produced by quaich.

Annotation of cLADs was taken from GSE76594 [29], and coordinates of cLADs were converted from hg19 to hg38 using liftover.

### 4.5. Genomic Regions with Altered Compaction

To select bins with the ratio *R* between the number of contacts within a 1 Mb vicinity and sum of all cis-contacts changed in Tr cells compared to normal karyotype, thresholds of 0.78 and 1.23 were used. Such thresholds correspond to 5% and 85% percentile of aneuploidy samples, respectively. To identify genomic regions with altered compaction, selected bins were merged using the *cluster* function from the bioframe package v0.4.1, allowing bins with either increased or decreased values to be merged only if the distance between them is equal to or less than 1 bin – *min_dist =* 50,000. The mean of the fold change of all the bins included into this cluster were assigned to each cluster. Clusters consisting of only one bin (no neighbors crossing the thresholds) were discarded. The aforementioned procedure was done for bins with increased and decreased *R* separately (clusters “UP” and “DOWN”, respectively).

### 4.6. Overlap with A/B Compartments

To assess the fraction of each cluster corresponding to A or B compartments, we used the *coverage* function from the bioframe package.

### 4.7. Location of Protein-Coding Genes

To check whether UP and DOWN clusters are enriched with protein-coding genes, we downloaded gene annotation from Gencode v43 for hg38 assembly. We calculated the number of intersections between clusters and genes using the *intersect* function from pybedtools v0.9.0. To check statistical significance of such a number, we used a shuffle test with 1000 iterations. Shuffles were created using the *shuffle* function from pybedtools v0.9.0.

### 4.8. Location of Housekeeping Genes

We used a previously published list of housekeeping genes [42] and used gene names to obtain coordinates from Gencode v43 for the hg38 assembly, as the original list was made for the hg19 assembly. Enrichment of housekeeping genes in UP and DOWN clusters was calculated the same way as for protein-coding genes.

### 4.9. Fluorescence In Situ Hybridization

Cell fixation, permeabilization and pre-hybridization treatments were performed according to a published protocol [43]. Briefly, cells were fixed in 4% PFA/ 1× PBS for 10 min at RT, washed three times in 0.05% Triton X-100/ 1× PBS and permeabilized in 0.5% Triton X-100/1× PBS for 10 min at RT. Next, coverslips with cells were incubated in 20% glycerol/1× PBS for an hour, permeabilized by several freeze–thaw cycles in liquid nitrogen and treated with 0.1 N HCl for 10 min at RT and 200 mkg/mkl RNAse for 30 min at 37 °C. Coverslips with cells were incubated in 50% formamide/2× SSC for an hour at RT just before hybridization.

FISH with commercial whole-chromosome probes (Kreatech FISH probes available from Leica Biosystems; Cat. #KBI-30018G, KBI-30021R and KBI-30022R for chr18, chr21 and chr22, respectively) and posthybridization washes were performed according to the manufacturer’s protocol (Kreatech FISH probes, Leica Biosystems, Wetzlar, Germany). Briefly, cells were denatured in 70% formamide/2× SSC for 15 min at 70 °C. Hybridization was performed at 37 °C for 16–18 h in a humid chamber. Coverslips were washed in 2× SSC for 2 min at RT, then in 0.4× SSC/0.3% NP-40 for 2 min at 72 °C and in 2× SSC/0.1% NP-40 for 2 min at RT. Cells were counterstained with DAPI and mounted in anti-fade medium (90% glycerol/25 mg/mL DABCO).

Cells were analyzed by the epifluorescent microscope system AxioVision and an EC Plan-NeoFluar ×100/1.3 oil objective (Zeiss, Oberkochen, Germany). Images were processed using Fiji software (NIH, ImageJ, version number 1.53q). For background reduction, the function “Background substraction” was used for each image. Nuclei and FISH signals were thresholded using Otsu’s and Yen’s method, respectively. Distances between FISH signals and nuclear lamina were counted by finding the shortest distance between the center of mass of each FISH signal and the edge of the nucleus, determined by DAPI staining. Collected data were normalized to the nuclear radius. Data were statistically analyzed using the SciPy Python library (v.1.10.1).

### 4.10. Reanalysis of Previously Published Hi-C Data from Tr21 iPSCs and iPSC-Derived NPCs

Raw Hi-C reads were downloaded from GSE185192. Replicates were merged, and Hi-C data were processed using the distiller-nf pipeline (https://github.com/open2c/distiller-nf (accessed on 1 June 2023)) in exactly the same way as original Hi-C data (see Section 4.4).

## Figures and Tables

**Figure 1 ijms-24-16044-f001:**
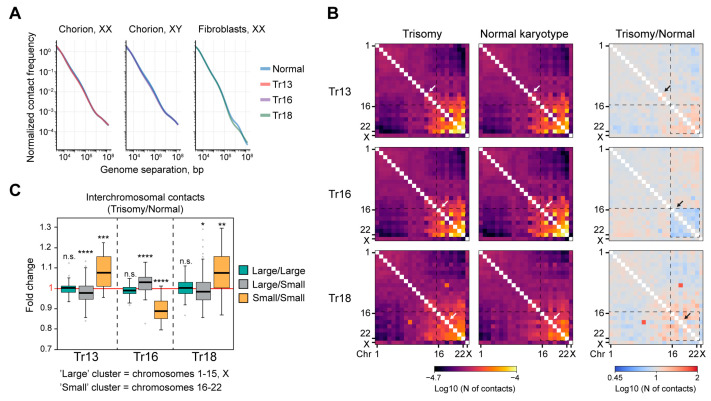
Trisomies affect contacts between large and small chromosomes. (**A**) *P_c_*(*s*) plots for normal and trisomic cells. Genetic background (XX, XY) is indicated. (**B**) Whole-genome maps on interchromosomal contacts. White and black arrows indicate Tr chromosomes. Dashed squares show “large” (with the exception of chrX) and “small” chromosome clusters. (**C**) Distributions of fold change of interchromosomal contact frequency (trisomy/normal cells). ****—*p* < 0.0001, ***—*p* < 0.001, **—*p* < 0.01, *—*p* < 0.05, n.s.—non-significant in a Mann–Whitney U-test.

**Figure 2 ijms-24-16044-f002:**
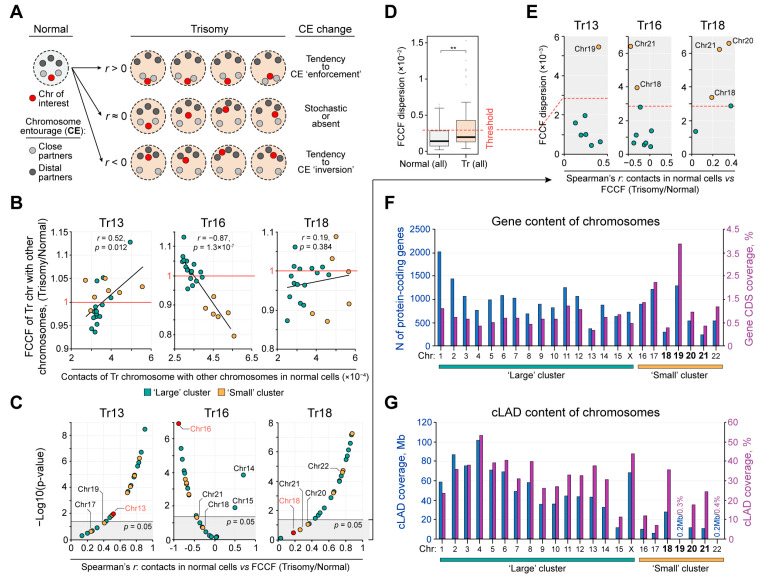
Chromosomes respond individually to the presence of an extra chromosome. (**A**) Interpretation of the chromosome entourage (CE) and its changes in trisomic cells. (**B**) The dependence between contact frequency of the Tr chromosome with other chromosomes in normal cells and contact frequency fold change (FCCF) (trisomy/normal). *r*—Spearman’s correlation coefficient. (**C**) Spearman’s correlation coefficient between contact frequency of a chromosome with other chromosomes in normal cells and its fold change (trisomy/normal). Ranged by value and significance. (**D**) Dispersion of contact number fold change (trisomy/normal) for all chromosomes in all trisomies (Tr) and in all pair-wise comparisons of normal cells. **—*p* < 0.01 in a Mann–Whitney U-test. (**E**) Dispersion of contact number fold change (trisomy/normal) for chromosomes with statistically non-significant *r* from panel (**C**). (**F**,**G**) Gene and cLAD content of chromosomes. CDS coverage is shown according to the UCSC browser, and cLAD coverage is shown according to ref. [29].

**Figure 3 ijms-24-16044-f003:**
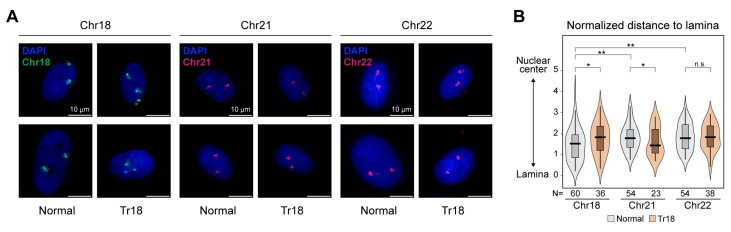
Chr18 and Chr21 change radial positions in Tr18 fibroblasts. (**A**) Representative examples of normal and Tr18 fibroblast nuclei. Scale bar—10 μm. (**B**) Distributions of distance to lamina for chr18, chr21 and chr22; normalized to the nuclear radius. For comparisons in normal fibroblasts and for comparison of ‘normal vs. Tr18′ for chr18 and chr22: **—*p* < 0.01, *—*p* < 0.05, n.s.—non-significant in a Mann–Whitney U-test; for comparison of ‘normal vs. Tr18’ for chr21: *—*p* < 0.05 in a Kolmogorov–Smirnov test.

**Figure 4 ijms-24-16044-f004:**
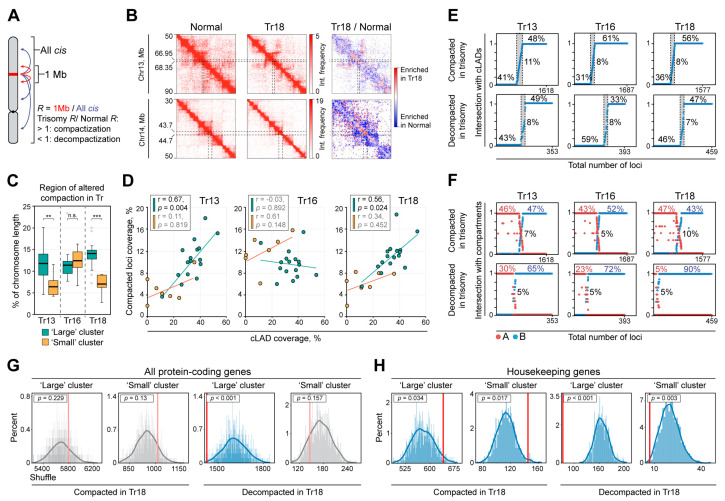
Trisomies alter chromatin compaction throughout the entire genome. (**A**) Scheme illustrating calculation of the *R* value. (**B**) Representative examples of loci compacted in Tr18 fibroblasts. (**C**) Percent of chromosome length covered by regions with altered chromatin compaction in Tr cells. ***—*p* < 0.001, **—*p* < 0.01, n.s.—non-significant in a Mann–Whitney U-test. (**D**) The dependence between chromosome coverage with compacted loci and cLADs. *r*—Spearman’s correlation coefficient, calculated for large and small chromosomes separately. (**E**) Loci with altered chromatin compaction ranged by the length fraction overlapped with cLADs. Percentage of loci with no overlap (0) and fully overlapped with cLADs (1) is indicated. (**F**) The same as panel (**E**), overlapping with A/B compartments. (**G**,**H**) Localization of all protein coding genes (**G**) and housekeeping genes (**H**) in loci with altered chromatin compaction.

**Figure 5 ijms-24-16044-f005:**
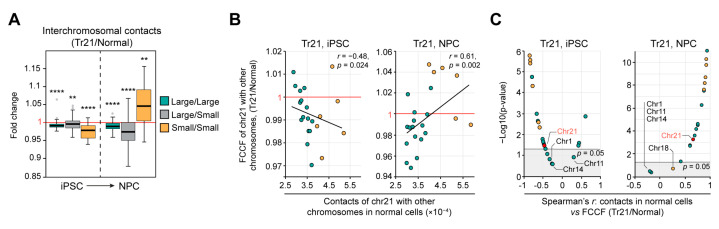
Chromosome behavior in Tr21 cells. (**A**) Distributions of fold change of interchromosomal number of contacts (Tr21/normal cells). ****—*p* < 0.0001, **—*p* < 0.01 in a Mann–Whitney U-test. (**B**) The dependence between contact frequency of chromosome 21 with other chromosomes in normal cells and contact frequency fold change (FCCF) (Tr21/normal). *r*—Spearman’s correlation coefficient. (**C**) Spearman’s correlation coefficient between contact frequency of a chromosome with other chromosomes in normal cells and its fold change (trisomy/normal). Ranged by value and significance.

## Data Availability

Raw and processed Hi-C data are available in the GEO repository under the accession number GSE237372.

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
