# Peer review of "Trisomies Reorganize Human 3D Genome"

_ijms, 2023, doi:10.3390/ijms242216044_

Round 1

Reviewer 1 Report (New Reviewer)

Comments and Suggestions for Authors

This is an interesting study, which is well described. The basic question the authors addressed as to how a presumed altered spatial organization of chromosomes in the nucleus due to trisomy for one chromosome may affect gene expression. The topic and the underlying hypothesis are original and certainly important for the field. The used methods are adequate and need no improvements. The conclusions are amply supported by the data and clearly discussed. The references are appropriate and the figures clear.

Only minor English editing is needed (for instance, "gene doze" should read as "gene dose").

Comments on the Quality of English Language

D ear Editor,

Thank you the opportunity to review this manuscript.

since part of the text is in red color, I take it that this manuscript has already undergone a round of reviewing and revision.  With a little language editing this manuscript will be fit to print.

Author Response

This is an interesting study, which is well described. The basic question the authors addressed as to how a presumed altered spatial organization of chromosomes in the nucleus due to trisomy for one chromosome may affect gene expression. The topic and the underlying hypothesis are original and certainly important for the field. The used methods are adequate and need no improvements. The conclusions are amply supported by the data and clearly discussed. The references are appropriate and the figures clear.

Only minor English editing is needed (for instance, "gene doze" should read as "gene dose").

REPLY

We thank the reviewer for positive feedback. In the revised version of the MS, we replaced "gene doze" with "gene dosage". In addition, the entire text was proofread by a cytogeneticist.

Reviewer 2 Report (New Reviewer)

Comments and Suggestions for Authors

The Authors present an interesting study of chromatin 3D structure in cells with autosomal trisomies using a high-throughput chromatin conformation capture technique (Hi-C).

First, the manuscript should be proofread and corrected by a native English speaker who is familiar with scientific terms, as the Authors have used words that have different meanings than desired or do not exist at all. See for example:

Page 2, line 57: "large arm" should be "long arm."

Page 2, line 59: "gene doze" should be "gene dosage"

Page 2, line 61: "supernumerical" should be "supernumerary".

There are methodological problems in the study design.

It is not correct to proceed with such a sophisticated study without knowing the karyotype of the cells being examined. The techniques used to ascertain the presence of trisomy (CGH-Array and FISH) are not adequate. Particularly for trisomy 13, we do not know whether it is a free or translocation trisomy, and this can greatly affect the results. To know the karyotype of a cell, a karyotype examination must be performed! The absence of a karyotype examination also has an impact on the possible interpretation of the presumed rearrangement between chromosomes 9 and 18 in fibroblasts with trisomy 18.

The karyotype problem also affects the cells used as controls. When the Authors write "As a control, we used chorion cells and fibroblasts with normal karyotype." do they mean cells with a normal karyotype or cells without genomic imbalances? These are two very different things, which may impact the results of this type of study.

Moreover, it would have been preferable to study cells with the same origin (all from chorionic villi or all from skin biopsies). In fact, as the Authors also write, the presence of the same extra chromosome could have different effects in different cell types. It is true that the Authors correctly used the same type of cells as controls with respect to the samples, but this does not solve the problem, as it does not allow generalizing conclusions about the results obtained in different trisomies.

It would be appropriate to provide more details on the 3D FISH analysis, as it is critical for all evaluations made on the chromosome to lamina distances. Incidentally, it does not seem to me to be reported which probes were used for the experiments! The text mentions "commercial whole-chromosome probes" but the images appear to be obtained with centromeric probes.

Some statements should be revised. See for example:

Page 2, lines 88-89: "Tr16, the most wide-spread trisomy in humans."

Page 10, section 4.2: The title (Karyotype analysis) is not appropriate because the section does not contain any karyotype data.

The same problem is also there in the caption of Supplementary Figure 1.

There is confusion between karyotype examination and CGH-Array and FISH.

Finally, as a general rule, all acronyms should be made explicit the first time they are used in the text.

Comments on the Quality of English Language

The manuscript should be proofread and corrected by a native English speaker who is familiar with scientific terms, as the Authors have used words that have different meanings than desired or do not exist at all. See for example:

Page 2, line 57: "large arm" should be "long arm."

Page 2, line 59: "gene doze" should be "gene dosage"

Page 2, line 61: "supernumerical" should be "supernumerary".

Author Response

(POINT 1) The Authors present an interesting study of chromatin 3D structure in cells with autosomal trisomies using a high-throughput chromatin conformation capture technique (Hi-C).

First, the manuscript should be proofread and corrected by a native English speaker who is familiar with scientific terms, as the Authors have used words that have different meanings than desired or do not exist at all. See for example:

Page 2, line 57: "large arm" should be "long arm."

Page 2, line 59: "gene doze" should be "gene dosage"

Page 2, line 61: "supernumerical" should be "supernumerary".

REPLY

In the revised version of the MS, we introduced these corrections and performed proofreading by a native English speaker.

(POINT 2) There are methodological problems in the study design.

It is not correct to proceed with such a sophisticated study without knowing the karyotype of the cells being examined. The techniques used to ascertain the presence of trisomy (CGH-Array and FISH) are not adequate. Particularly for trisomy 13, we do not know whether it is a free or translocation trisomy, and this can greatly affect the results. To know the karyotype of a cell, a karyotype examination must be performed! The absence of a karyotype examination also has an impact on the possible interpretation of the presumed rearrangement between chromosomes 9 and 18 in fibroblasts with trisomy 18.

REPLY

The medical center that provided us with the research material does not conduct classical cytogenetic analysis. Therefore, in the revised version of the MS, we rephrased the text, eliminating the term “karyotype” (highlighted in red in the text):

The presence of extra chromosomes was revealed by the array comparative genome hybridization (aCGH) technique for the Tr13 and Tr16, and by the chromosomal analysis and fluorescence in situ hybridization (FISH) with the whole-chromosome probes for the Tr18 (Supplementary Fig. 1A).

Indeed, CGH-Array does not distinguish free from translocated trisomy. However, a translocation between chr13 and another chromosome would inevitably lead to a sharp increase in the contact probability between the two chromosomes (similar to the pair chr9-chr18 in fibroblasts). Hi-C is a very sensitive technique for detecting chromosomal rearrangements (see e.g. 10.1186/s13059-017-1253-8). We do not detect an abnormally high interaction frequency between chr13 and any other chromosome in Hi-C data from Tr13 chorion (see Figure 1B). This clearly indicates that Tr13 cells carry free extra chr13.

We suggest that chr9-chr18 rearrangement is represented by a short translocation rather than chromosome fusion, because FISH imaging of chr9 with telomeric probes did not detected the presence of splitted chr9 (both telomeric probes are presented within the same chromosome, and we did not find chromosomes with only one signal). Besides, chr18 behavior is similar in three non-related cell types analyzed in this work. We thus believe that the 18q-9q rearrangement did not substantially affect the contacts of chr18 with other chromosomes.

(POINT 3) The karyotype problem also affects the cells used as controls. When the Authors write "As a control, we used chorion cells and fibroblasts with normal karyotype." do they mean cells with a normal karyotype or cells without genomic imbalances? These are two very different things, which may impact the results of this type of study.

REPLY

To clarify this statement, we modified the text as follows:

“Chorion cells without genomic imbalances detectable by CGH-array hybridization were isolated from two male and three female donors…”

(POINT 4) Moreover, it would have been preferable to study cells with the same origin (all from chorionic villi or all from skin biopsies). In fact, as the Authors also write, the presence of the same extra chromosome could have different effects in different cell types. It is true that the Authors correctly used the same type of cells as controls with respect to the samples, but this does not solve the problem, as it does not allow generalizing conclusions about the results obtained in different trisomies.

REPLY

We thought that it would be important to find out if the effects of trisomies are or are not not cell type-specific. To this end, we reasoned that this work would benefit from the Hi-C analysis of non-related cell types.

(POINT 5) It would be appropriate to provide more details on the 3D FISH analysis, as it is critical for all evaluations made on the chromosome to lamina distances. Incidentally, it does not seem to me to be reported which probes were used for the experiments! The text mentions "commercial whole-chromosome probes" but the images appear to be obtained with centromeric probes.

REPLY

Following the reviewer’s suggestions, we expanded the description of the 3D FISH analysis (highlighted in red in the revised MS):

Coverslips with cells were incubated in 50% formamide/2xSSC for an hour at RT just before hybridization.

FISH with commercial whole-chromosome probes (Kreatech FISH probes available from Leica Biosystems; Cat. #KBI-30018G, KBI-30021R and KBI-30022R for chr18, chr21 and chr22, respectively) and posthybridization washes were performed according to manufacturer protocol (Kreatech FISH probes, Leica Biosystem). Briefly, cells were denaturated in 70% formamide/2xSSC for 15 minutes at 70oC. Hybridization was performed at 37oC for 16-18 hours in a humid-chamber. Coverslips were washed in 2xSSC for 2 minutes at RT, then in 0.4xSSC/0.3% NP-40 for 2 minutes at 72oC and in 2xSSC/0.1% NP-40 for 2 minutes at RT. Cells were counterstained with DAPI and mounted in anti-fade medium (90% glycerol/ 25 mg/ml DABCO).”

“Collected data were normalized to the nuclear radius. Data were statistically analyzed using SciPy Python library (v.1.10.1).”

As indicated in the above sentences, we definitely used whole-chromosome Kreatech FISH probes available from Leica Biosystems (Cat. #KBI-30018G, KBI-30021R and KBI-30022R).

(POINT 6) Some statements should be revised. See for example:

Page 2, lines 88-89: "Tr16, the most wide-spread trisomy in humans."

Page 10, section 4.2: The title (Karyotype analysis) is not appropriate because the section does not contain any karyotype data.

The same problem is also there in the caption of Supplementary Figure 1.

REPLY

Following the reviewer’s suggestion, we corrected the mentioned statements in the revised MS:

“Tr16, the most frequent trisomy in the analysis of spontaneous abortion material”

Section 4.2: “Karyotype analysis” has been replaced with “Microarray analysis”

A new caption for the Supplementary Figure 1: “(A) Microarray analysis in Tr chorion cells (left) and FISH analysis in Tr fibroblasts (right).”

(POINT 7) There is confusion between karyotype examination and CGH-Array and FISH.

REPLY

In the revised MS, we clarified this and replaced “karyotype analysis” with more appropriate descriptions (see POINT 2).

(POINT 8) Finally, as a general rule, all acronyms should be made explicit the first time they are used in the text.

REPLY

We fixed this issue in the revised version.

Reviewer 3 Report (New Reviewer)

Comments and Suggestions for Authors

Authors report a HI-C study on trisomic human tissues (+13, +16, or +18) and show that +13 and +16 change the 3D-structure of the genome.

Comments:

-          Abstract – please explain abbreviation LAD

-          Abstract – explain what “human donor chorion cells” are or change just to “human chorion cells”

-          Abstract/ Discussion: in abstract is stated “The presence of extra 28 chromosomes 13 and 16, but not 18, results…” how does that fir to statement in line 359, ff “Moreover, we found that specifically Tr13 and Tr18, but not Tr16 are characterized by a strong dependency between the proportion of loci with altered compaction and LAD content of a chromosome”?

-          Introduction: line 45 – add that there are also multiple copies of Y-chr. possible; also there is the possibility of double trisomies in const. cases.

-          Introduction: line 59 – correct doze to dosage

-          Introduction: line 74 – one letter s is in red – change to black

-          Introduction: line 81 – add the papers showing that also presence of a small supernunmerary marker chromosome (sSMC) can change the 3D-structure

-          Results: line 91/92 – it may be ok to show free trisomy 18 by aCGH, definitely for trisomy 13 it is necessary to check on chromosomes if there is a free trisomy 13 or a trisomy based on Robertsonian translocation. If this is not possible any more in the used case make a clear statement in abstract, results, discussion that a Robertsonian translocation cannot be excluded in the Patau syndrome case studied.

-          Results: as only one tissue per patient was studied the results obtained are very limited; it must be also mentioned in abstract, results, discussion that for trisomy 13 and 16 chorion was studied and for trisomy 18 fibroblasts – the difference of trisomies 13 and 16 influencing the 3D-structure and trisomy 18 not influencing the 3D-structure can just be due to tissue specific differences observed – this reduces the meaningfulness of this study a lot

-          Results point 2.1  -  as it has been shown that human chromosomes (in sperm) follow two kinds of influences in their special rearrangement (size and gene content) see Manvelyan et al. in Mol Cytogenet – these points need to be included and not just grouping as done by the authors – same for point 2.2. – the results concerning LAD are published before in mentioned Manvelyan paper and others before.

-          Legend of Fig. 2 – some words are in red – change to black

-          Results – can it be seen from the data if extra chromosomes colocalize – from sSMC data this is to be expected.

-          Results – point 2.4 – previously published data must be presented here acc. to tissues studied – see comment above for chorion and fibroblast data described here. Data from IPS cells should be also discussed separately as they are far away from real human body cells.

-          Results: check also for interphase FISH-studies on trisomic cells and nuclear architecture – a quick search revealed at least one such paper in const. cases PMID: 28396582 (trisomy 21) and one in leukemia (trisomy 8) PMID: 19639225, one for solid tumors (trisomy 7) PMID: 30909073.
Also check for more HIC-papers like https://doi.org/10.1101/2022.11.04.515239;

-          Discussion: check statement of first sentence – chromatin includes DNA and (histone) proteins – most likely authors talk about DNA part?

-          Discussion: paragraph starting in line 340 – the LAD idea combined with large chromosomes is nice – but acc. to most papers the lethality of large chromosome trisomies is correlated with gene content – this is nicely supported by the fact that trisomy 19 is absolutely lethal and trisomy 18 (chromosomes 18 and 19 of almost same size) is not lethal in >50% of the cases…. Please avoid or massively rewrite that paragraph.

-          Discussion from line 369 onwards  – some words are in red – change to black

-          MM part . 4.2 – if an aCGH is done this cannot be called ‘Karyotype analyses’! it is a CNV analyses – not more and not less!

-          MM part  - 4.10 – headline needs to read as Fluorecsence in situ …

-          MM part needs a 4.11. – explain here the literature research mentioned in results!

Author Response

Authors report a HI-C study on trisomic human tissues (+13, +16, or +18) and show that +13 and +16 change the 3D-structure of the genome.

Comments:

(POINT 1) Abstract – please explain abbreviation LAD

REPLY

We fixed this issue in the revised version.

(POINT 2) Abstract – explain what “human donor chorion cells” are or change just to “human chorion cells”

REPLY

In the revised version, we replaced “human donor chorion cells” with “human chorion cells”.

(POINT 3) Abstract/ Discussion: in abstract is stated “The presence of extra chromosomes 13 and 16, but not 18, results…” how does that fit to statement in line 359, ff “Moreover, we found that specifically Tr13 and Tr18, but not Tr16 are characterized by a strong dependency between the proportion of loci with altered compaction and LAD content of a chromosome”?

REPLY

In Abstract, we stated that Tr13 and Tr16 alter interchromosomal (trans) contacts. In the Results, we describe the influence of Tr13 and Tr18 on intrachromosomal (cis) contact patterns, specifically in relation to LAD content. These two types of contacts (trans and cis, correspondingly) are substantially different.

(POINT 4) Introduction: line 45 – add that there are also multiple copies of Y-chr. possible; also there is the possibility of double trisomies in const. cases.

REPLY

We added this information in the revised version of the MS

(POINT 5) Introduction: line 59 – correct doze to dosage

REPLY

This has been corrected in the revised version of the MS

(POINT 6) Introduction: line 74 – one letter s is in red – change to black

REPLY

This has been corrected in the revised version of the MS

(POINT 7) Introduction: line 81 – add the papers showing that also presence of a small supernumerary marker chromosome (sSMC) can change the 3D-structure

REPLY

The following sentence was added to the introduction section of the revised MS:

"Even the presence of a small supernumerary marker chromosome (sSMC) has been reported to influence the arrangement of other chromosomes within the cell nucleus (ref PMID: 26616419)”.

(POINT 8) Results: line 91/92 – it may be ok to show free trisomy 18 by aCGH, definitely for trisomy 13 it is necessary to check on chromosomes if there is a free trisomy 13 or a trisomy based on Robertsonian translocation. If this is not possible any more in the used case make a clear statement in abstract, results, discussion that a Robertsonian translocation cannot be excluded in the Patau syndrome case studied.

REPLY

Indeed, CGH-Array does not discriminate free trisomy from the translocation one. However, translocation between chr13 and other chromosome would inevitably lead to drastic increase in contact probability between the two chromosomes. Hi-C is a very sensitive technique for the detection of chromosome rearrangements (for example, see 10.1186/s13059-017-1253-8). We do not detect abnormally high interaction frequency between chr13 and any other chromosome in Hi-C data from Tr13 chorion (see Figure 1B). This clearly indicates that Tr13 cells used in our study carry free extra chr13.

(POINT 9) Results: as only one tissue per patient was studied the results obtained are very limited; it must be also mentioned in abstract, results, discussion that for trisomy 13 and 16 chorion was studied and for trisomy 18 fibroblasts – the difference of trisomies 13 and 16 influencing the 3D-structure and trisomy 18 not influencing the 3D-structure can just be due to tissue specific differences observed – this reduces the meaningfulness of this study a lot

REPLY

We should note that all studied trisomies influence the 3D genome structure, but in different ways. For instance, Tr13 and Tr16, but not Tr18 alter interchromosomal contacts. Tr13 and Tr18 (obtained from different cell types, specifically from chorion and fibroblasts), but not Tr16, are characterized by a strong dependency between the proportion of loci with altered compaction and LAD content of a chromosome. Moreover, chr18 demonstrates similar behavior in different trisomies in different cell types (see paragraph 2.4). Therefore, we believe that the identified effects are not cell type-specific.

(POINT 10) Results point 2.1  -  as it has been shown that human chromosomes (in sperm) follow two kinds of influences in their special rearrangement (size and gene content) see Manvelyan et al. in Mol Cytogenet – these points need to be included and not just grouping as done by the authors – same for point 2.2. – the results concerning LAD are published before in mentioned Manvelyan paper and others before.

REPLY

Concerning chromosome clustering: we realize that the partitioning of chromosomes into “small” and “large” cluster is arbitrary. In fact, there is no sharp drop neither in chromosome size nor in key features such as gene coverage. However, a tendency to spatial clustering of small and large chromosomes is obvious, although the boundary between the clusters is blurred and is located somewhere between chr12 and 16 (“transition zone”). We thus believe that the exact localization of the boundary is not critical for the analysis and might depend on cell type and/or physiological state of the cells.

In Manvelyan et al. in Mol Cytogenet (similar to numerous other studies, including ones cited in our MS) radial positions of chromosomes are described. However, LADs are not studied in Manvelyan et al. We thus refrain from citing this paper as not relevant to our work.

(POINT 11) Legend of Fig. 2 – some words are in red – change to black

REPLY

This has been corrected in the revised MS.

(POINT 12) Results – can it be seen from the data if extra chromosomes colocalize – from sSMC data this is to be expected.

REPLY

The FISH analysis performed in our study did not reveal any colocalization of extra chromosomes.

(POINT 13) Results – point 2.4 – previously published data must be presented here acc. to tissues studied – see comment above for chorion and fibroblast data described here. Data from IPS cells should be also discussed separately as they are far away from real human body cells.

REPLY

We are not quite sure that we understood the reviewer's suggestion correctly. However, in the original MS, we present the data exactly as the reviewer suggest: Sections 2.1-2.3 describe observations from chorion cells and fibroblasts, and Section 2.4 present results from iPSCs and iPSCs-derived NPCs.

(POINT 14) Results: check also for interphase FISH-studies on trisomic cells and nuclear architecture – a quick search revealed at least one such paper in const. cases PMID: 28396582 (trisomy 21) and one in leukemia (trisomy 8) PMID: 19639225, one for solid tumors (trisomy 7) PMID: 30909073.
Also check for more HIC-papers like https://doi.org/10.1101/2022.11.04.515239;

REPLY

PMID: 28396582 does not provide information about large-scale architecture of chromosomal interactions in trisomic cells because the authors performed FISH analysis only for chr21 and found “…that two copies of maternal chromosomes resulting from meiotic nondisjunction had a higher tendency to form an adjacent pair and were located relatively distant from the nuclear membrane, suggesting the conserved interaction between these homologous chromosomes”. Our data do not allow us to discriminate interactions established by maternal and paternal copies of chromosomes, and we do not take it into account during data analysis. Therefore, discussion of the relative positions of copies of extra chromosomes is beyond the scope of our work.

PMID: 30909073 is already cited in the original version of the MS.

https://doi.org/10.1101/2022.11.04.515239 - this preprint does not contain any biological information related to trisomies. The authors just show that their variant of Hi-C protocol is suitable for the detection of trisomies (page 7 in the preprint: “As shown in Fig. 5a bottom, both trisomies were identified, demonstrating that liCHi-C can be used for genome-wide detection of CNVs and to scan breakpoints from primary patient tissues without the need for a reference.”). Conventional Hi-C is also a high-sensitive technique for the detection of chromosomal abnormalities (for example, see Harewood et al. 2017, 10.1186/s13059-017-1253-8). We thus thing that citing of this preprint is not relevant.

(POINT 15) Discussion: check statement of first sentence – chromatin includes DNA and (histone) proteins – most likely authors talk about DNA part?

REPLY

The first statement of the Discussion indicates the nuclear volume occupied by chromatin fibers, not just DNA per se. (See Ou, H. D.; Phan, S.; Deerinck, T. J.; Thor, A.; Ellisman, M. H.; O'Shea, C. C., ChromEMT: Visualizing 3D chromatin structure and compaction in interphase and mitotic cells. Science 2017, 357, (6349)).

(POINT 16) Discussion: paragraph starting in line 340 – the LAD idea combined with large chromosomes is nice – but acc. to most papers the lethality of large chromosome trisomies is correlated with gene content – this is nicely supported by the fact that trisomy 19 is absolutely lethal and trisomy 18 (chromosomes 18 and 19 of almost same size) is not lethal in >50% of the cases…. Please avoid or massively rewrite that paragraph.

REPLY

First, we have to note that neither chr18 nor chr19 belong to large chromosomes. While chr18 is gene-poor and LAD-rich (in this regard chr18 is similar to large chromosomes), it is obviously small (and preferentially interact with other small chromosomes; see Figure 1 and numerous published Hi-C data from different cell types and biological conditions). Thus, our suggestion about potential relationships between LAD content of large chromosomes and pathological effects of trisomies is not relevant for the pair of chr18-chr19. To avoid confusion, we modified the text as follows:

Thus, chromosomes containing a large proportion of LADs (predominantly chromosomes from the large cluster) may compete for the localization at the nuclear periphery that is important for the maintenance of a repressed state of genes located in LADs [34].

(POINT 17) Discussion from line 369 onwards  – some words are in red – change to black

REPLY

This has been corrected in the revised version of the MS

(POINT 18) MM part . 4.2 – if an aCGH is done this cannot be called ‘Karyotype analyses’! it is a CNV analyses – not more and not less!

REPLY

In the revised version of the MS, we reworded the text excluding the term "karyotype":

The presence of extra chromosomes was revealed by the array comparative genome hybridization (aCGH) technique for the Tr13 and Tr16, and by the chromosomal analysis and fluorescence in situ hybridization (FISH) with the whole-chromosome probes for the Tr18 (Supplementary Fig. 1A).

(POINT 19) MM part  - 4.10 – headline needs to read as Fluorecsence in situ …

REPLY

This has been corrected in the revised MS

(POINT 20) MM part needs a 4.11. – explain here the literature research mentioned in results!

REPLY

A new paragraph 4.11 has been added to the revised version:

“Raw Hi-C reads were downloaded from GSE185192. Replicates were merged and Hi-C data were processed using the distiller-nf pipeline (https://github.com/open2c/distiller-nf) in exactly the same way as original Hi-C data (see paragraph 4.4).”

Round 2

Reviewer 2 Report (New Reviewer)

Comments and Suggestions for Authors

The authors did a good job of reviewing, responding to all the comments made. The manuscript seems to me to be much improved.

Author Response

The authors did a good job of reviewing, responding to all the comments made. The manuscript seems to me to be much improved.

REPLY

We thank the reviewer for positive feedback.

Reviewer 3 Report (New Reviewer)

Comments and Suggestions for Authors

Thanks for the work on the paper

Still no clear are following points

(POINT 3) Authors state in their reply that what this referee referred to in abstract and text are two different things – still authors need to avoid misunderstandings and this point is still not clear for the reader – please revise and make clear what is meant at which point in abstract and text and how these data can be understood together without any (seemingly) contradictions.

(POINT 7) here it might by also interesting to refer to PMID: 22413994 where different influence of hetero- and euchromatic sSMCs was shown.

(POINT 8) please include your reply in the paper – it is a god argument that the studied case most likely had a free and no translocation trisomy 13.

(POINT 9) please include your reply in the paper

(POINT 10) The answer is not clear – Manvelyan paper is just an example how general literature, to which the authors also refer see influence of gene density and chromosome size. Why to use other, new groups for LAD evaluation?

(POINT 13) What is requested is to show own and previous results in a Table and aligned by cell type. This would support authors claim that all cell types behave alike in case of trisomy. This si not proven yet from this reviewer’s point of view, and authors could provide that.

(POINT 16) is rephrased here
Discussion: paragraph starting in line 340 – the LAD idea combined with large chromosomes is nice – but acc. to most papers the lethality of large chromosome trisomies is correlated with their large gene content compared to smaller ones with less gene content.
This is nicely supported by the fact that trisomy 19 is absolutely lethal and trisomy 18 (chromosomes 18 and 19 of almost same size) is not lethal in >50% of the cases…. Please avoid or massively rewrite that paragraph, as it seems to say that larger chromosomes are lethal not due to gene content but to LADs. This cannot be (the whole) true(th), as shown by lethality of small chromosome 19 and lethality of all chromosomes, irrespective of size and LADs apart from #13, #18 and #21, being relatively gene poor.

(POINT 20) please explain how papers were identified and selected – inclusion/ exclusion criteria for studies found by Pubmed search?

Author Response

Still no clear are following points

(POINT 3) Authors state in their reply that what this referee referred to in abstract and text are two different things – still authors need to avoid misunderstandings and this point is still not clear for the reader – please revise and make clear what is meant at which point in abstract and text and how these data can be understood together without any (seemingly) contradictions.

REPLY

To make it more clear, we incorporated the following sentence into the Abstract of the revised version of the MS:

“For trisomy 13 and 18, but not for trisomy 16, the proportion of compacted loci on a chromosome is correlated with LAD content.”

Also, we modified the following sentence in the Abstract section to bring it in line with data presented in Figure 1B and 1C:

“The presence of extra chromosomes results in systematic changes of contact frequencies between small and large chromosomes.”

(POINT 7) here it might by also interesting to refer to PMID:22413994 where different influence of hetero- and euchromatic sSMCs was shown.

REPLY

In the revised version of the MS, we cited PMID:22413994 in the Intro section:

“…even though sSMCs tend to colocalize with their corresponding sister chromosomes [25].”

(POINT 8) please include your reply in the paper – it is a god argument that the studied case most likely had a free and no translocation trisomy 13.

REPLY

We included this information into the revised version of the MS:

Notably, aCGH does not discriminate free trisomy from the translocation one. It is of particular importance for Tr13 because chr13 is frequently involved in Robertsonian translocations. However, translocation between chr13 and other chromosomes would inevitably lead to drastic increase in contact probability between the two chromosomes. We do not detect abnormally high interaction frequency between chr13 and any other chromosome in Hi-C data from Tr13 chorion (see Figure 1B). This clearly indicates that Tr13 cells used in this study carry free extra chr13.”

(POINT 9) please include your reply in the paper

REPLY

We have to note that all parts (listed below) are already present in the initial version of the MS:

Part 1: We should note that all studied trisomies influence the 3D genome structure, but in different ways.

This statement is one of the main conclusions of our work. In this regard, in one form or another, we repeat it several times during the presentation of the results, as well as in the Discussion section.

Part 2: For instance, Tr13 and Tr16, but not Tr18 alter interchromosomal contacts.

Actually, this is not true, and we apologize for this inaccuracy. As shown in Figure 1B and 1C, all trisomies alter interchromosomal interactions. See lines 154-160 of the initial MS:

“However, we found that the presence of  extra chromosomes induces moderate yet statistically significant and opposite changes in contacts between small and large chromosomes (Figure 1C). In Tr13 and Tr18, we observed further segregation of the large and small chromosome clusters accompanied by an increase in contact frequency between small chromosomes, while Tr16 cells are characterized by a partial intermingling of the clusters and a pronounced decrease in interactions of small chromosomes.”

Part 3: Tr13 and Tr18 (obtained from different cell types, specifically from chorion and fibroblasts), but not Tr16, are characterized by a strong dependency between the proportion of loci with altered compaction and LAD content of a chromosome.

This is stated in lines 265-267 of the MS (note the fixed typo in line 265: Tr16 has been corrected to Tr13):

“In the Tr13 and Tr18 cells, we found a strong correlation between the cLAD content and the proportion of compacted regions on large, but not small chromosomes (Figure 4D).”

Part 4: Moreover, chr18 demonstrates similar behavior in different trisomies in different cell types (see paragraph 2.4). Therefore, we believe that the identified effects are not cell type-specific.

This is stated in lines 316 -329 of the MS:

Thus, chr18 is the only chromosome from the “small” cluster whose interactions not change or change stochastically in three out of four analyzed trisomies (Tr16, Tr18 and Tr21) in unrelated cell types (chorionic cells, fibroblasts and NPCs, respectively).”

(POINT 10) The answer is not clear – Manvelyan paper is just an example how general literature, to which the authors also refer see influence of gene density and chromosome size. Why to use other, new groups for LAD evaluation?

REPLY

As we stated previously, we realize that the partitioning of chromosomes into “small” and “large” cluster is arbitrary. However, we applied this partitioning to underlie the effects observed in our data and we strongly believe that this way is appropriate. Besides, we note that chromosome size, LAD coverage and gene density are tightly linked features of chromosomes: large chromosomes are characterized by relatively small gene density and high LAD coverage. This is a textbook knowledge which does not require citation of any particular paper.

(POINT 13) What is requested is to show own and previous results in a Table and aligned by cell type. This would support authors claim that all cell types behave alike in case of trisomy. This si not proven yet from this reviewer’s point of view, and authors could provide that.

REPLY

All the obtained results and their interpretation are presented in Figures and precisely described in the Results section. We really do not see how another graphical item (table) would improve our argumentation. A table might have been a better way to present the data if several dozen cell types had been studied. But given that only five cell types have been studied, presenting the results in Figures seems reasonable.

(POINT 16) is rephrased here

Discussion: paragraph starting in line 340 – the LAD idea combined with large chromosomes is nice – but acc. to most papers the lethality of large chromosome trisomies is correlated with their large gene content compared to smaller ones with less gene content.

This is nicely supported by the fact that trisomy 19 is absolutely lethal and trisomy 18 (chromosomes 18 and 19 of almost same size) is not lethal in >50% of the cases…. Please avoid or massively rewrite that paragraph, as it seems to say that larger chromosomes are lethal not due to gene content but to LADs. This cannot be (the whole) true(th), as shown by lethality of small chromosome 19 and lethality of all chromosomes, irrespective of size and LADs apart from #13, #18 and #21, being relatively gene poor.

REPLY

Our data (see Figure 4) suggest that LAD content is linked in a certain way to changes in intrachromosomal contact frequencies in trisomic cells. Based on this experimental observation, we present a hypothesis on a potential relationship between LAD content of large chromosomes and pathological consequences of large chromosomes trisomies. Large chromosomes contain a large number of LADs; for instance, while LAD coverage of chr18 and chr2 is similar (about 35%), chr2 contains about 90 Mb of LADs, and chr18 contains only about 15 Mb of LADs. This difference allows to look at the lethality of large chromosomes trisomies from another angle. However, we do not question the fact that the gene number may be important as well. To make this clear we added the following sentence in the revised version of the MS:

“We should also note that a large number (not density) of genes on large chromosomes per se could contribute to embryonic lethality of large trisomies.”

(POINT 20) please explain how papers were identified and selected – inclusion/ exclusion criteria for studies found by Pubmed search?

REPLY

We based our selection on our evaluation of quality of data presented and on the data relevance for our research.

This manuscript is a resubmission of an earlier submission. The following is a list of the peer review reports and author responses from that submission.

Round 1

Reviewer 1 Report

Comments and Suggestions for Authors

Zhegalova el al. presented an interesting study on 3D aberration among different trisomies. However, I have following questions:

(1)    The descriptions need to be rewritten throughput the manuscript. For instance, in Page 2 Lines 95-97, the authors stated that “Chorion cells with normal karyotype were isolated from two male and three female donors, Tr13 and Tr16 chorion were obtained from one donor (female and male, respectively).” For the cells with Trisomy 13 and Trisomy 16, were they originated from different individuals? It was saying from one donor but in the bracket, it was saying female and male, respectively. In addition, “Primary Tr18 and normal fibroblasts were isolated from one female donor.” from the same page, did the authors mean that the patient with T18 in mosaicism? Or primary T18 and normal fibroblasts came from different individuals as well? They made me confused.

(2)    Page 3, line 109 “sequenced with 11.5-150 millions of paired-end reads, and 6.1-93.3 millions of unique”. After reviewing the supp Table S1, each lib from case (Trisomy cell) only obtained ~15 million read-pairs, while each control lib got significantly higher number of read-pairs. Would this affect the analysis such as comparison. In addition, with such limited number of read-pairs, the claimed resolution (50kb) is questionable.

(3)    From Figure 1B, the authors stated that the results indicated more frequent contacts between “small” chromosomes and with a cutoff of chromosomes 1 to 15. However, it was only reflected by the Trisomy/Normal figure in Trisomy 16 data (less contact in “small” chromosome). But the results from the other figures might not reflect this claim. It seems that the increased contacts have involved other chromosomes such as chromosomes 14 and 15 as well.

(4)    Page 3 Lines 133 “this is likely caused by a local rearrangement between 18q and pericentromeric region of 9q.” As t(11;22) translocation is already known to impact on the radial location of the derivative chromosomes in the nucleus, the translocation between chromosomes 9q and 18q is believed to have a similar impact. If so, the inter-chromosomal contacts between other chromosomes and the chromosomes 18 (including the two derivative chromosomes) would be affected as well. Therefore, in order to exclude such factor, the authors should try to perform analysis in a T18 cell line without other gross chromosomal abnormalities identified.

(5)    Different cell types were used in this study but none of the data were generated from different cell types but with the same trisomy. In addition, as some studies demonstrated the boundaries of topologically associated domains (TADs) would be conserved across different cell types, it is also important to review whether there were aberrations of TADs as well.

Comments on the Quality of English Language

Although most contents are readable, like what I mentioned, there are sentences causing confusion. Please revise accordingly to make the statement clearly.

Author Response

Zhegalova el al. presented an interesting study on 3D aberration among different trisomies. However, I have following questions:

(1) The descriptions need to be rewritten throughput the manuscript. For instance, in Page 2 Lines 95-97, the authors stated that “Chorion cells with normal karyotype were isolated from two male and three female donors, Tr13 and Tr16 chorion were obtained from one donor (female and male, respectively).” For the cells with Trisomy 13 and Trisomy 16, were they originated from different individuals? It was saying from one donor but in the bracket, it was saying female and male, respectively. In addition, “Primary Tr18 and normal fibroblasts were isolated from one female donor.” from the same page, did the authors mean that the patient with T18 in mosaicism? Or primary T18 and normal fibroblasts came from different individuals as well? They made me confused.

Reply:

In the revised version of the MS, we modified this paragraph as follows:

“…Tr13 and Tr16 chorion were obtained from one female and one male donor, respectively.

Primary Tr18 and normal fibroblasts were isolated from female donors (one donor for the Tr18 and one donor for the normal fibroblasts).

(2) Page 3, line 109 “sequenced with 11.5-150 millions of paired-end reads, and 6.1-93.3 millions of unique”. After reviewing the supp Table S1, each lib from case (Trisomy cell) only obtained ~15 million read-pairs, while each control lib got significantly higher number of read-pairs. Would this affect the analysis such as comparison. In addition, with such limited number of read-pairs, the claimed resolution (50kb) is questionable.

Reply:

As stated in the Methods section, all libraries were subsampled to the same number of reads using cooltools random-sample prior the analysis. Thus, we believe that different sequencing depth should not affect the results. Note that technical replicates (G and Y in Supplementary Table 1) were combined before the analysis, and the ~15 million read-pairs are cis interactions. This correspond to 250 contacts per 50-kb bin, on average, that is sufficient for the analysis of local chromatin compaction (at the scale of LADs). A/B compartments were determined at 500-kb resolution.

(3) From Figure 1B, the authors stated that the results indicated more frequent contacts between “small” chromosomes and with a cutoff of chromosomes 1 to 15. However, it was only reflected by the Trisomy/Normal figure in Trisomy 16 data (less contact in “small” chromosome). But the results from the other figures might not reflect this claim. It seems that the increased contacts have involved other chromosomes such as chromosomes 14 and 15 as well.

Reply:

We realize that the splitting of chromosomes into “small” and “large” cluster is arbitrary. In fact, there is no sharp drop neither in chromosome size nor in key features such as gene coverage. However, a tendency to spatial clustering of chromosomes is obvious, although the boundary between the clusters is blurred and is located somewhere between chr12 and 16 (“transition zone”). We thus believe that the exact localization of the boundary is not critical for the analysis and might depend on cell type and/or physiological state of the cells.

(4) Page 3 Lines 133 “this is likely caused by a local rearrangement between 18q and pericentromeric region of 9q.” As t(11;22) translocation is already known to impact on the radial location of the derivative chromosomes in the nucleus, the translocation between chromosomes 9q and 18q is believed to have a similar impact. If so, the inter-chromosomal contacts between other chromosomes and the chromosomes 18 (including the two derivative chromosomes) would be affected as well. Therefore, in order to exclude such factor, the authors should try to perform analysis in a T18 cell line without other gross chromosomal abnormalities identified.

Reply:

We suggest that chr9-chr18 rearrangement is represented by a short translocation rather than chromosome fusion, because FISH imaging of chr9 with telomeric probes did not detected the presence of splitted chr9 (both telomeric probes are presented within the same chromosome, and we did not find chromosomes with only one signal). Besides, chr18 behavior is similar in three non-related cell types analyzed in this work. We thus believe that the 18q-9q rearrangement did not substantially affect the contacts of chr18 with other chromosomes. To clarify this, in the revised version of the MS, we added a new Supplementary Figure 1C showing FISH imaging of chr9 and modified the text as follows:

FISH imaging of chr9 with subtelomeric probes did not show the presence of splitted chr9, since both probes are readily detected within the same chromosome (Supplementary Fig. 1C). This suggests that the rearrangement did not result in the formation of chimeric chromosome between chr9 and chr18. Nonetheless, chr9 was excluded from all downstream comparisons of normal and Tr18 fibroblasts.

(5) Different cell types were used in this study but none of the data were generated from different cell types but with the same trisomy. In addition, as some studies demonstrated the boundaries of topologically associated domains (TADs) would be conserved across different cell types, it is also important to review whether there were aberrations of TADs as well.

Reply:

Indeed, we did not perform Hi-C experiments in different cell types with the same trisomy. However, in the initial version of the MS, we reanalyzed previously published data from iPSCs and NPCs carrying extra copy of chr21 (Meharena et al. 2022, Cell stem cell; see Figure 5 in the MS).

In this work, we are mainly focused on chromosome behavior in aneuploid cells. A limited analysis of local chromatin compaction has been performed to investigate a potential connection between changes in interchromosomal contacts and chromosome association with the nuclear lamina. Reliable detection of TAD boundaries and assessment of their strength is out of the scope of this study. It requires more deep sequencing of the libraries and implies some additional assays such as RNA-seq and epigenetic profiling since the TAD boundary calling per se is not informative for understanding the relationships between the presence of an extra chromosome and genome disfunction. Thus, we believe that TAD (and loop) profile examination requires a separate study.

(6) Although most contents are readable, like what I mentioned, there are sentences causing confusion. Please revise accordingly to make the statement clearly.

Reply:

We modified confused sentences in the revised version.

Reviewer 2 Report

Comments and Suggestions for Authors

The topic of 3D genome organization is an interesting, relatively new field in need of additional information. Despite my excitement for the topic, I had concerns regarding the presentation of the study results and discussion, as summarized below:

 1.       In several places throughout the manuscript, the authors incorrectly use cytogenetic terms or information. For example, the authors use triploidy (which refers specifically to a condition with 69 chromosomes) as a “substitute” for trisomy for an individual chromosome (page 4, line 173).  On page 1, lines 44-46, the authors note that “tetrasomy, pentasomy and even hexasomy of autosomes are also present in some types of cancer and in patients with mental disorders . . .”.  While this is correct for cancer, it is not correct for patients who have constitutional findings.  Partial imbalances involving autosomes can be seen as constitutional anomalies, but not full tetrasomy, etc among liveborns.

2. On page 2 and at other points in the manuscript, the authors use the term “linked”.  Given that linked and linkage have specific definitions in the field of genetics, this term is confusing and not fully accurate in the context it is presented. An alternative term to use could be “associated”.

3. It appears that only 1 specimen was evaluated for each of the trisomy conditions. How do the authors know if the single case they evaluated is representative of most cases having this aneuploidy condition?  At least two, and ideally, at least 3 different specimens should be evaluated for each trisomy.

 4.      On page 3, lines 131-134, the authors indicate that for the trisomy 18 fibroblasts, the chromosome 18 showed “an abnormally strong contact with chr9”. They further stated that this observation was likely caused by a local rearrangement between 18q and the pericentromeric region of 9q and elected to exclude chromosome 9 from the comparisons.  Did the authors complete a chromosomal study to determine if this specimen (or any of the other specimens) had a translocation that might impact its genome organization?  All of the cells used should be karyotyped to confirm if they have any structural chromosomal findings.  Also, given that chromosome 9 has a large heterochromatic pericentromeric region and chromosome 18 has a large heterochromatin content, could this be a meaningful observation? How do they know it is atypical for chromosome 18 if they only looked at one specimen?

 5.      Also, on pages 2 and 3 the authors state that trisomy 18 and normal fibroblasts were isolated from one female donor.  Did that person have constitutional mosaicism?  They note that these comparisons were performed within the same genetic background (XX or XY). Was the comparison really the same genome-wide background (mosaic) or did the just have the same sex chromosome complement?

6.      The authors use the term “inversion” to describe the coalescent pattern they observe. In the field of cytogenetics, inversion refers to a specific type of structural chromosomal abnormality. The use of inversion to describe the interphase nucleus chromatin placement/organization described is confusing.

 7.      On page 5, last paragraph, the authors describe differential localization of heterochromatic compared to euchromatic regions. This observation is not new, but has been previously reported by several investigators, who should be cited.

 8.      The authors’ criteria for categorizing a chromosome as “large” or “small” is unclear.  For example, there is a significant size difference between chromosomes 1, 11, and 14, yet each of these is clustered as “large”.  This categorization seems quite arbitrary.  At least adding a medium category seems indicated, but one could also use chromosome specific length values to complete a more quantitative comparison.

 9.      On page 9 the authors make interesting points about the nucleolus and lamina associations.  Given this fact, they may wish to consider analyzing the acrocentric chromosomes (which are involved in nucleolus formation) separately from the other chromosomes.

 10. The description of the Hi-C library preparation seems long compared to the other methodology descriptions.  Also, on page 13, the authors present an atypical method for performing FISH that involves several freeze-thaw cycles.  Why was this done?  What is the purpose of this approach? Could this atypical approach lead to DNA damage?

Comments on the Quality of English Language

There are some language concerns throughout the document (pronoun use, etc). The greatest language concern relates to the author’s description of  cytogenetic findings and their use of words that could lead to confusion, such as “inversion” rather than coalescence or congregation. 

Author Response

The topic of 3D genome organization is an interesting, relatively new field in need of additional information. Despite my excitement for the topic, I had concerns regarding the presentation of the study results and discussion, as summarized below:

  1. In several places throughout the manuscript, the authors incorrectly use cytogenetic terms or information. For example, the authors use triploidy (which refers specifically to a condition with 69 chromosomes) as a “substitute” for trisomy for an individual chromosome (page 4, line 173). On page 1, lines 44-46, the authors note that “tetrasomy, pentasomy and even hexasomy of autosomes are also present in some types of cancer and in patients with mental disorders . . .”. While this is correct for cancer, it is not correct for patients who have constitutional findings. Partial imbalances involving autosomes can be seen as constitutional anomalies, but not full tetrasomy, etc among liveborns.

Reply:

We thank the reviewer for these comments. We modified the text as follows:

Page 1: “However, tetrasomy, pentasomy and even hexasomy of autosomes are also present in some types of cancers [1] and multiple copies of large chromosome fragments are found in patients with mental disorders [2]. Multiple copies of the X chromosome are associated with severe development defects [3].

Page 4: “This is also true for the extra chromosomes in Tr cells (Figure 2B).

  1. On page 2 and at other points in the manuscript, the authors use the term “linked”. Given that linked and linkage have specific definitions in the field of genetics, this term is confusing and not fully accurate in the context it is presented. An alternative term to use could be “associated”.

Reply:

We fixed this inaccuracy throughout the text.

  1. It appears that only 1 specimen was evaluated for each of the trisomy conditions. How do the authors know if the single case they evaluated is representative of most cases having this aneuploidy condition? At least two, and ideally, at least 3 different specimens should be evaluated for each trisomy.

Reply:

Definitely, all the observations made in this work should be tested in future on a larger number of donors in different cell types. It would also benefit from the study of other and more rare trisomies. For instance, the presence of extra chromosome 2 is of particular interest, because it occasionally presents in missed miscarriages, according to the statistics of Research Centre for Medical Genetics (Moscow, Russia; personal communication). However, within the framework of this research, the following observations allow us to think that the obtained results are reliable:

(1) Coherent behavior of chr13 and chr18 in Tr13 chorion cells and Tr18 fibroblasts, respectively. Both these chromosomes are gene-poor and LAD-rich, and they demonstrate qualitatively similar changes in contact profiles (see Figures 2B, 2C, 4C and 4D).

(2) The presence of extra chr13 and extra chr18 induces similar changes in interchromosomal contacts of other chromosomes (see Figure 1C).

(3) The presence of chr16 which is gene-rich and LAD-poor induces opposite changes in interchromosomal contact profiles compared to extra chr13 and chr18. Moreover, chr16 per se changes its contact frequencies with other chromosomes in an opposite manner as well.

(4) Chr18 changes its contact frequency with other chromosomes in a similar way in non-related cell types (Tr16 chorionic cells, Tr18 fibroblasts and Tr21 NPCs).

Taken together, these observations made on cells of different origin, derived from different donors fit the common logic indicating the role of LAD and gene content in orchestrating chromosome behavior in aneuploid cells.

  1. On page 3, lines 131-134, the authors indicate that for the trisomy 18 fibroblasts, the chromosome 18 showed “an abnormally strong contact with chr9”. They further stated that this observation was likely caused by a local rearrangement between 18q and the pericentromeric region of 9q and elected to exclude chromosome 9 from the comparisons. Did the authors complete a chromosomal study to determine if this specimen (or any of the other specimens) had a translocation that might impact its genome organization? All of the cells used should be karyotyped to confirm if they have any structural chromosomal findings. Also, given that chromosome 9 has a large heterochromatic pericentromeric region and chromosome 18 has a large heterochromatin content, could this be a meaningful observation? How do they know it is atypical for chromosome 18 if they only looked at one specimen?

Reply:

We suggest that chr9-chr18 rearrangement is represented by a short translocation rather than chromosome fusion, because FISH imaging of chr9 with telomeric probes did not detected the presence of splitted chr9 (both telomeric probes are presented within the same chromosome, and we did not find chromosomes with only one signal). Besides, chr18 behavior is similar in three non-related cell types analyzed in this work. We thus believe that the 18q-9q rearrangement did not substantially affect the contacts of chr18 with other chromosomes. To clarify it in the revised version of the MS, we added a new Supplementary Figure 1C showing FISH imaging of chr9 and modified the text as follows:

FISH imaging of chr9 with subtelomeric probes did not show the presence of splitted chr9, since both probes are readily detected within the same chromosome (Supplementary Fig. 1C). This suggests that the rearrangement did not result in the formation of chimeric chromosome between chr9 and chr18. Nonetheless, chr9 was excluded from all downstream comparisons of normal and Tr18 fibroblasts.

  1. Also, on pages 2 and 3 the authors state that trisomy 18 and normal fibroblasts were isolated from one female donor. Did that person have constitutional mosaicism? They note that these comparisons were performed within the same genetic background (XX or XY). Was the comparison really the same genome-wide background (mosaic) or did the just have the same sex chromosome complement?

Reply:

We thank the reviewer for this comment. Tr18 and normal fibroblasts were isolated from different donors of the same genetic background (XX). For clarity, we modified the text as follows:

Primary Tr18 and normal fibroblasts were isolated from female donors (one donor for the Tr18 and one donor for the normal fibroblasts).

  1. The authors use the term “inversion” to describe the coalescent pattern they observe. In the field of cytogenetics, inversion refers to a specific type of structural chromosomal abnormality. The use of inversion to describe the interphase nucleus chromatin placement/organization described is confusing.

Reply:

To avoid confusion, in the revised version of the MS, we have replaced the term “inversion” with “reversal”.

  1. On page 5, last paragraph, the authors describe differential localization of heterochromatic compared to euchromatic regions. This observation is not new, but has been previously reported by several investigators, who should be cited.

Reply:

In the revised version of the MS, we cited a classic review by Cremer and Cremer (2001) describing principles of chromosome positioning within the nucleus. Also, we slightly modified this sentence as follows:

In agreement with previous studies (reviewed in [26]), we observed a difference in radial positions of these chromosomes in normal cells (Figure 3A)…

  1. The authors’ criteria for categorizing a chromosome as “large” or “small” is unclear. For example, there is a significant size difference between chromosomes 1, 11, and 14, yet each of these is clustered as “large”. This categorization seems quite arbitrary. At least adding a medium category seems indicated, but one could also use chromosome specific length values to complete a more quantitative comparison.

Reply:

Partitioning of chromosomes into “small” and “large” clusters is arbitrary. In fact, there is no sharp drop neither in chromosome size nor in key features such as gene coverage. However, a tendency to spatial clustering of chromosomes is obvious, although the boundary between the clusters is blurred and resides somewhere between chr12 and 16 (“transition zone”). Therefore we believe, that the exact localization of the boundary is not critical for the analysis and might depend on cell type and/or physiological state of the cells.

  1. On page 9 the authors make interesting points about the nucleolus and lamina associations. Given this fact, they may wish to consider analyzing the acrocentric chromosomes (which are involved in nucleolus formation) separately from the other chromosomes.

Reply:

This is an interesting point that we have analyzed prior the submission. However, we did not find any statistically significant in contacts specifically between chromosomes containing nucleolus organizer regions.

  1. The description of the Hi-C library preparation seems long compared to the other methodology descriptions. Also, on page 13, the authors present an atypical method for performing FISH that involves several freeze-thaw cycles. Why was this done? What is the purpose of this approach? Could this atypical approach lead to DNA damage?

Reply:

Since the Hi-C technique is a primary source of data in this work, we believe that Hi-C description should be detailed (to date, a large number of variants of the Hi-C protocol have been published, and there is no “gold standard”).

We used FISH protocol which has been previously published by the Cremer lab (ref. 41). Several freeze-thaw cycles are required for the soft permeabilization of cells, and glycerol serves as a cryoprotective agent preventing DNA damage.

There are some language concerns throughout the document (pronoun use, etc). The greatest language concern relates to the author’s description of cytogenetic findings and their use of words that could lead to confusion, such as “inversion” rather than coalescence or congregation.

Reply:

In the revised version of the MS, we attempted to fix all the language issues.

Reviewer 3 Report

Comments and Suggestions for Authors

This is a very interesting and well executed study. 

I have only a few remarks:

- please perfom a full spelling check as there a still a few semantic mistakes

- could you comment on the choice of your tissues, more specific why did you use chorionic cells and fibroblasts?

Comments on the Quality of English Language

Overall the English language is fine, there are some minor semantic mistakes, e.g.

- line 205: 'Surprisingly short list of chromosomes...' should be 'A surprisingly...'

- line 214: 'Thus, LAD and gene content may be among determinants...' should be '..may be among the..'

The paper would benefit from proofreading by a native speaker.

Author Response

This is a very interesting and well executed study.

I have only a few remarks:

  1. Please perfom a full spelling check as there a still a few semantic mistakes

Reply:

We thank the reviewer for positive feedback. The MS has been proofread.

  1. Could you comment on the choice of your tissues, more specific why did you use chorionic cells and fibroblasts?

Reply:

Chorionic cells was the only primary cell type available from abortion tissues derived from donors, and primary fibroblasts were the only Tr18 cell type available from the collection of Research Centre for Medical Genetics (Moscow, Russia).

  1. Overall the English language is fine, there are some minor semantic mistakes, e.g.

3.1. Line 205: 'Surprisingly short list of chromosomes...' should be 'A surprisingly...'

Reply:

The error has been fixed.

3.2. Line 214: 'Thus, LAD and gene content may be among determinants...' should be '..may be among the..'

Reply:

The error has been fixed.

3.3. The paper would benefit from proofreading by a native speaker.

Reply:

English spelling has been checked throughout the text.

Round 2

Reviewer 1 Report

Comments and Suggestions for Authors

The authors have addressed my concerns, I don't have further questions.